# Oleanolic Acid Acetate Exerts Anti-Inflammatory Activity via IKKα/β Suppression in TLR3-Mediated NF-κB Activation

**DOI:** 10.3390/molecules24214002

**Published:** 2019-11-05

**Authors:** Hyung Jin Lim, Hyun-Jae Jang, Mi Hwa Kim, Soyoung Lee, Seung Woong Lee, Seung-Jae Lee, Mun-Chual Rho

**Affiliations:** 1Immunoregulatory Material Research Center, Korea Research Institute of Bioscience and Biotechnology, 181 Ipsin-gil, Jeongeup-si 56212, Korea; lhjin@kribb.re.kr (H.J.L.); seung_jae99@naver.com (M.H.K.); sylee@kribb.re.kr (S.L.); lswdoc@kribb.re.kr (S.W.L.); 2Department of Bioactive Material Sciences, Chonbuk National University, Jeonju-si 54896, Korea; 3Natural Medicine Research Center, Korea Research Institute of Bioscience and Biotechnology, 30 Yeongudanji-ro, Cheongju-si 28116, Korea; water815@kribb.re.kr

**Keywords:** *Vigna angularis*, oleanolic acid acetate, toll-like receptors, inflammatory cytokines, IκB kinase α/β

## Abstract

Oleanolic acid acetate (OAA), a major triterpenoid compound of *Vigna angularis* (azuki bean, *V. angularis*), has been shown to downregulate inflammatory responses in macrophages. Here, we show the molecular basis for the effect of OAA on Toll-like receptor (TLR) downstream signaling. OAA treatment significantly inhibited the secretion of embryonic alkaline phosphatase (SEAP) induced by polyinosinic acid (poly(I), TLR3 ligand) in a dose-dependent manner and without cytotoxicity in THP1-XBlue cells. In addition, OAA downregulated the gene expression of poly(I) induced pro-inflammatory cytokines and chemokines genes such as MCP-1, IL-1β, IL-8, VCAM-1 and ICAM-1. Furthermore, we found that the inhibition activity of OAA was accompanied by decreased activation of not only nuclear factor-kappa B (NF-κB) signaling but also mitogen-activated protein kinase (MAPK) signaling upon stimulation with the TLR3 agonist. Interestingly, the interaction of OAA with IκB kinase α/β (IKKα/β) strongly attenuated the production of certain proteins and inflammatory cytokines in the TLR3 signaling pathway, such as nuclear factor of kappa light polypeptide gene enhancer in B-cells inhibitor, alpha (IkBα), extracellular regulated kinases (ERK), and p38, in an in vitro model. The action of OAA was regulated by TLR3, demonstrating that TLR3 plays a critical role in mediating the physiologically-relevant anti-inflammatory action of OAA and that the interaction with IKKα/β is modulated through TLR3. These results reveal new insight into the understanding of the regulatory mechanisms of the downstream TLR3 signaling pathway and consequent inflammatory responses that are involved in the development and progression of inflammatory diseases.

## 1. Introduction

Toll-like receptors (TLRs) are essential receptors that recognize and respond to pathogen-associated molecular patterns (PAMPs), and are pathogen-recognition proteins that play crucial roles in detecting pathogens and initiating inflammatory responses [1,2]. The TLR family in humans consists of ten members (TLRs 1 to 10), and the cytoplasmic portion of TLR family proteins, known as the Toll/IL-1 receptor-like (TIR) domain, is similar to that of interleukin 1 receptor (IL-1R) family proteins. In contrast, the extracellular portions of TLR family proteins are structurally unrelated, and each TLR discriminates between specific patterns of bacterial components [2]. The recognition of specific patterns of microbial components by TLRs activates intracellular signaling pathways by recruiting TIR domain adaptors conserved among all TLR family members, including myeloid differentiation factor (MyD)88, MyD88 adapter-like (Mal) TIR domain containing adaptor protein (TIRAP/MAL), TIR domain containing adapter inducing interferon-β (TRIF), and TRIF related adaptor molecule (TRAM), which, consequently, leads to alterations in the expression of inflammatory cytokine and chemokine genes [3,4].

In general, the TLR signaling pathways consist of two cascades: MyD88-dependent and TRIF-dependent (MyD88-independent) [5]. The MyD88-dependent pathway mediates the production of proinflammatory cytokines such as tumor necrosis factor-α (TNF-α) and interleukin-8 (IL-8) in response to all TLR ligands, whereas the TRIF-dependent pathway is necessary for the induction of type I interferons (IFNs) mediated by TLR3 and TLR4. The recruitment of MyD88 by activated TLRs leads to the autophosphorylation of IL-1R-related kinase (IRAK), which results in the translocation of NF-κB to the nucleus via the tumor necrosis factor receptor-related factor 6 (TRAF6)/transforming growth factor β-activated kinase-1 (TAK-1) signaling pathway. Concomitantly, c-Jun and c-Fos, members of the activator protein-1 (AP-1) complex, are activated via the TRAF6/MAPK signaling pathway, leading to the production of proinflammatory cytokines via NF-κB [2,6].

Many investigators have focused on searching for small molecule inhibitors from natural resources to modulate TLR-specific responses [7]. To date, numerous studies have shown that phytochemicals, such as flavonoid compounds and polyphenolic compounds, exert anti-inflammatory effects by inhibiting TLR agonist-induced NF-κB activation [8].

The inflammatory response is one of the defense mechanisms against viral and bacterial infections. However, excessive production of inflammatory cytokines or chemokines by overactive inflammatory responses disrupts the immune balance and leads to hypersensitivity reactions or chronic diseases such as rheumatoid arthritis, atherosclerosis, and lung fibrosis [9,10,11]. Therefore, controlling the TLR signaling pathway could be a preventive or therapeutic approach to attenuate excessive inflammation.

*Vigna angularis* (*V. angularis*) is a common legume used in the daily diet or as a plant-based medicine in Asia. Oleanolic acid acetate (OAA), the major *V. angularis* oleanane-type triterpenoid, is mainly present in the hull. Many studies have identified and described the various pharmacological activities of OAA, such as its antiosteoporotic and antiallergy activity, and treatment effects in rheumatoid arthritis and atopic dermatitis [12,13]. However, the effect of OAA on TLR activation or the mechanism by which OAA exerts these effects has not yet been studied. In this study, we show that OAA inhibits TLR3-induced inflammatory responses by focusing on NF-κB signaling pathways and proinflammatory cytokines.

## 2. Materials and Methods

### 2.1. Preparation of OAA from V. angularis

OAA was purified from *V. angularis* as previously described [12]. In brief, *V. angularis* was obtained from herbal medicine stores (Jeongeup, Korea). The pulverized *V. angularis* (10 kg) was treated twice with 95% ethanol (EtOH) solvent at 70 °C. The EtOH extracts were filtered and concentrated under reduced pressure. The EtOH extracts were suspended in distilled water and successively separated into ethyl acetate (EtOAc) solvent. OAA was isolated by chromatography on silica gel (Merck, Darmstadt, Germany) using a step gradient *n*-hexane:EtOAc solvent system (100:1, 80:1, 60:1, 40:1, 20:1, 10:1, 1:1; each 1 L, *v*/*v*). Biotinylated OAA was prepared in our laboratory (see Appendix A).

### 2.2. Cell Culture

Human leukemia monocytic cell line, THP1 cells, were purchased from ATCC (Manassas, VA, USA). Cells were maintained in RPMI 1640 medium supplemented with 10% fetal bovine serum (FBS), 100 U/mL penicillin, and 100 µg/mL streptomycin. THP1-XBlue cells were purchased from Invivogen (San Diego, CA, USA) and maintained in RPMI 1640 medium supplemented with 10% FBS, 200 µg/mL zeocin, 100 µg/mL blasticidin, 100 U/mL penicillin, and 100 µg/mL streptomycin at 37 °C in a 5% CO_2_ incubator.

### 2.3. Toll-Like Receptor (TLR) Activation Assay

THP1-XBlue cells are transfected with a reporter plasmid containing the secretion of embryonic alkaline phosphatase (SEAP) under the control of an NF-κB and AP-1 inducible promoter. The activation of TLR3 was determined by quantifying secreted SEAP. Briefly, cells were seeded at a plating density of 2 × 10^5^ cells per well, and cells were treated with OAA for 1 h before being incubated with polyinosinic:polycytidylic acid [poly(I:C)] and polyinosinic acid [poly(I)] (Sigma-Aldrich, St Louis, MO, USA). After 18 h of incubation at 37 °C in a 5% CO_2_ atmosphere, 20 µL of cell suspension was added to a new 96-well plate and mixed with 200 µL of QUANTI-Blue colorimetric assay reagent (Invivogen, San Diego, CA, USA) to detect SEAP. After incubation for 1 h for color development at 37 °C, quantitative readings were taken at 655 nm using a microplate reader.

### 2.4. Cell Viability

Cell viability was determined with an EZ-Cytox Cell viability assay kit (Daeil Lab Service, Seoul, Korea), which contains highly sensitive water-soluble tetrazolium salt (WST). THP1-XBlue cells were seeded at a plating density of 5 × 10^4^ cells per well, and after 3 h, the cells were treated with OAA at 5 to 50 µM. Following culture with OAA for 48 h, WST assay reagent (10 µL/well) was added and incubated for 4 h at 37 °C. The absorbance of the samples at 450 nm was measured against a background control using a microplate reader. The percentage of viable cells in each treatment condition was determined relative to the negative control.

### 2.5. Western Blot Analysis

Total cell lysates were extracted using cell lysis buffer (Cell Signaling Technology, Beverly, MA, USA) containing a protease and phosphatase inhibitor cocktail (Thermo Scientific, Rockford, IL, USA). Western blotting was performed as described previously [14]. Monocyte nuclear extracts were prepared with NE-PER Nuclear Extraction Reagents according to the manufacturer’s instructions (Thermo Scientific, Waltham, MA, USA). NF-κB p65 (ab7970) was obtained from Abcam Inc. (Cambridge, MA, USA), p-IKKα/β (Ser176/180), p-p44/42 MAPK (ERK1/2) (Thr202/Tyr204), p-p38 MAPK (Thr180/Tyr182), p-JNK (Thr183/Tyr185), p-TAK1 (Ser412), and p-IRF3 (Ser396) antibodies were purchased from Cell Signaling Technology Inc. (Beverly, MA, USA). NF-κB p50 (4D1), p-IκB (B-9), IκB (C-21), p-c-Jun (Ser63/73), p-c-Fos (Ser374), and lamin B antibodies were purchased from Santa Cruz Biotechnology Inc. The anti-β-actin antibody and secondary antibodies were purchased from Sigma-Aldrich Ltd. (St Louis, MO, USA).

### 2.6. RNA Isolation, Cdna Synthesis, and Quantitative Real-Time PCR

Total RNA was extracted from total cell lysates using a PureLink RNA Mini kit (Invitrogen, San Diego, CA, USA) according to the manufacturer’s instructions. First-strand cDNA was synthesized from 0.5 µg/mL total RNA using a Maxime RT PreMix Kit (iNtRON Biotechnology, Sungnam, Korea). Real-time PCR was performed on a Step One Plus Real-Time PCR System using TaqMan Real-Time PCR Master Mix (Applied BioSystems, Foster City, CA, USA). The analysis of relative mRNA levels of IL-1β, IL-8, MCP-1, and TNF-α was performed using TaqMan primers: IL-1β, Hs01555410; IL-8, Hs00174103; MCP-1, Hs00234140; TNF-α, and Hs00174128. Relative mRNA expression levels were quantified using the threshold cycle (Ct) value, and the Ct values were normalized using the human 18S rRNA gene as an endogenous reference.

### 2.7. Immunoprecipitation and Biotin–Streptavidin Pull-Down

According to the manufacturer’s instructions (Biotinylated Protein Interaction Pull-Down Kit, Thermo Scientific, Waltham, MA, USA), immunoprecipitation (IP) was performed on cell lysates. Briefly, cells were lysed in IP lysis buffer (Thermo Scientific, Waltham, MA, USA) for 30 min on ice, and then cell lysates were centrifuged at 13,000 rpm for 10 min at 4 °C. Agarose–streptavidin (50 μL) was transferred to fresh tubes and washed three times with binding buffer (25 mM Tris-HCl, 0.15 M NaCl, pH 7.0). After cell lysates (500 μg) were treated with biotin-labeled OAA (OAA-biotin) or OAA for 1 or 6 h, they were added to the agarose–streptavidin and incubated with rotation overnight at 4 °C. After the beads were washed three times with wash buffer (PBS with 0.01% Tween 20), they were incubated for 5 min at room temperature with elution buffer (50 μL). The immunoprecipitated proteins were immunoblotted with anti-Myd88, anti-TRAF6, anti-TAK1, anti-IKKα/β, anti-NF-κB p50, and anti-Biotin antibodies.

### 2.8. Statistical Analysis

Statistical analyses were performed 3 times for all experiments. All quantitative results are presented as means ± standard deviations (SD). Statistical analyses were performed using Prism 5 software (GraphPad Software, San Diego, CA) and statistical significance was determined by one-way ANOVA followed by Turkey’s test for multiple comparisons (* *p* < 0.05).

## 3. Results

### 3.1. The Effect of OAA on TLR Activation in Human Monocytes

In response to viral infections, each TLR interacts with different combinations of TLR domain adaptor proteins, leading to the activation of intracellular signaling pathways. Among the transcription factors activated by TLRs, NF-κB and AP-1, which regulate the inflammatory response, are the most common transcription factors in the TLR-mediated signal transduction pathway. To investigate whether TLR activation is modulated by OAA (Figure 1A), we investigated NF-κB and AP-1 reporter activity by using a SEAP reporter system. Poly(I), double-stranded RNA, and its synthetic analog, poly(I:C), are well-known activators of TLR3 [15]. The viability of OAA-treated cells was determined using a WST-1 assay, and OAA cytotoxicity was not observed at 3 to 100 μM concentrations (Figure 1B). Thus, we found that the TLR3 inhibition activity was not affected by OAA cytotoxicity. To examine the effect of OAA on TLR3 activation, we stimulated THP1-XBlue cells with poly(I:C) or poly(I) (10 μg/mL) in the presence or absence of OAA. As shown in Figure 1C, SEAP activity was dramatically increased by poly(I:C) or poly(I) treatment; OAA significantly reduced SEAP activity in a dose-dependent manner. Since OAA more effectively suppressed poly(I)- rather than poly(I:C)-induced TLR3 activation, we further investigated the effects of OAA on the target molecules involved in poly(I)-induced TLR3 activation (Figure 1C).

### 3.2. OAA Inhibits the TLR3-Mediated mRNA Expression of Proinflammatory Cytokines, Chemokines, and Proadhesive Molecules

Activation of the TLR3 signaling pathways initiates the expression of NF-κB and AP-1 target genes, including interleukin-1β (IL-1β), IL-8, monocyte chemoattractant protein 1 (MCP-1), and TNF-α. Accordingly, we investigated whether OAA inhibits the expression of inflammatory cytokine, chemokine, and adhesion molecule genes by inhibiting NF-κB and AP-1 using real-time quantitative PCR experiments. As shown in Figure 2, poly(I)-induced mRNA expression levels of MCP-1, IL-1β, IL-8, vascular cell adhesion molecule 1 (VCAM-1), and intercellular cell adhesion molecule 1 (ICAM-1) were significantly reduced by treatment with OAA (Figure 2), suggesting that the anti-inflammatory effect of OAA on proinflammatory molecule expression may be regulated via blockade of the TLR3 signaling pathway.

### 3.3. OAA Inhibits TLR3-Mediated Signaling

As shown in Figure 1 and Figure 2, OAA treatment inhibited the TLR3 activation-induced mRNA expression of proinflammatory mediators and transcription factors, such as NF-κB and AP-1. Thus, we examined whether OAA regulates transcriptional activation by analyzing the protein levels of NF-κB and AP-1 in nuclear extracts. As shown in Figure 3A, OAA treatment reduced the levels of the p65 subunit of the NF-κB complex and of the AP-1 component p-c-Fos in the TLR3 cascade activated by poly(I). This finding indicates that NF-κB and AP-1 translocation into the nucleus is inhibited by OAA treatment resulting in an anti-inflammatory effect. In the early stage of NF-κB translocation, IκB in NF-κB complexes is phosphorylated by IκB kinase (IKK), consisting of IKKα and IKKβ, resulting in its subsequent degradation [16]. We further examined the effect of OAA on the phosphorylation of IKKα/β and IκB. As shown in Figure 3B, OAA treatment significantly reduced the phosphorylation of IKKα/β, and inhibited IκB degradation. However, the activation of TAK1, which leads to the transcriptional activation of NF-κB and AP-1, was not affected by OAA treatment [17]. Thus, IKKα/β may be a target of OAA, involved in the inhibition of the nuclear translocation of the transcription factor NF-κB. Activation of AP-1, which is a dimeric complex of Fos and Jun proteins, is mainly associated with MAPK (ERK, JNK, and p38) signaling cascades [18,19]. To investigate whether OAA is involved in the MAPK pathway, we evaluated the effects of OAA on ERK, c-Jun N-terminal kinase (JNK), and p38 activation in the presence of poly(I). As shown in Figure 3C, OAA significantly decreased ERK phosphorylation; however, JNK and p38 phosphorylation was not inhibited by OAA treatment at concentrations of 10 to 60 μM. Consistent with the data in Figure 3A, the ERK MAPK-induced localization of c-Fos was blocked by OAA treatment. These findings suggest that OAA treatment inhibits AP-1 transcriptional activation via ERK MAPK signaling rather than JNK and p38 MAPK signaling. In contrast to that of other TLRs, the TRIF-dependent pathway of TLR3 involves the phosphorylation and dimerization of the IFN regulatory factor 3 (IRF3) transcription factor to induce the expression of the IFN-β gene [20]. As shown in Figure 3D, OAA treatment did not diminish the phosphorylation of IRF3. These results suggest that the anti-inflammatory activity of OAA may involve the TLR3/MyD88-dependent signaling pathway but not the TRIF-dependent pathway.

### 3.4. OAA Binds to IKKα/β Molecules

In Figure 3, we showed that IKKα/β activity in the poly(I)-stimulated TLR3 signaling pathway was inhibited by OAA treatment. To further investigate the possibility that OAA may directly interact with IKKα/β or its upstream regulatory proteins, we synthesized biotinylated OAA (biotin-OAA) (Figure 4) and performed pull-down experiments using biotin-OAA. Based on the heteronuclear multiple bond correlation (HMBC) between the biotin-terminal protons H-1′′ [δH 3.33 (m, H-1a′′) and 3.19 (m, H-1b′′)] and carbonyl carbon C-28 (δC 178.5), biotin was linked to the carboxyl group of OAA in the biotin-OAA chemical structure (see Appendix A). After cell extracts were incubated with biotin-OAA and OAA, the bound proteins were immunoprecipitated using streptavidin agarose beads. Precipitated proteins were resolved by SDS-PAGE, followed by immunoblotting with anti-biotin or specific antibodies. As shown in Figure 4A, NF-κB p50 and IKKα/β were successfully pulled down with biotin-OAA, suggesting that OAA interacts directly with the upstream molecule IKKα/β rather than with NF-κB. In addition, the interaction with IKKα/β was competitively reduced by cotreatment with biotin-OAA and OAA. However, biotin-OAA did not interact with TAK1, as did Myd88 and TRAF6 (data not shown), which are upstream of TAK1. Consistent with the results in Figure 3B, OAA did not affect proteins upstream of TAK1 in the poly(I)-induced TLR3 cascade in human monocytes, but was directly involved in the inhibition of IKKα/β activity resulting in the suppression of TLR3 signaling (Figure 4A). Taken together, these results demonstrate that OAA inhibits poly(I)-induced TLR3 signaling by specifically interacting with IKKα/β (Figure 4B).

## 4. Discussion

A common feature of TLR signaling is the activation of NF-κB through the TIR domain. The TLR signaling process involves MyD88 complex formation with the TIR region. The C-terminus of MyD88 complexes with TLR, and the IRAK-2 complexes with IRAK. IRAK-1 activates TRAF6 and NF-κB-induced kinase (NIK), which subsequently activates IKK (IκB kinase) resulting in the accumulation of activated IκB and NF-κB. Activated NF-κB induces the activity of nucleic acids and transcription enzymes and the expression of cytokine and nuclear genes [21,22]. After the recognition of dsRNA by TLR3 via the leucine-rich repeat motif, the adaptor protein TRIF is recruited to TLR3 to activate the transcription factors IRF3 and NF-κB and subsequent production of inflammatory cytokines, such as IL-6 and IL-12 [23]. The excessive production of inflammatory mediators results in an overactive immune response, which can lead to the worsening of various human diseases such as colitis, pancreatitis, rheumatoid arthritis, and asthma. The activation of NF-κB mainly occurs via the classic pathway, in which the IKK complex regulates the phosphorylation of the inhibitory molecule IκB-α. In the inactive state, NF-κB resides in the cytoplasm bound to IκB-α. In the active state, IκB-α is phosphorylated and degraded by ubiquitination. The free NF-κB; therefore, translocates to the nucleus and binds to the promoter region of target genes to control inflammation, immunity, cell proliferation, and apoptosis [24].

OAA, (a triterpenoid compound) a derivative of OA, is known to have therapeutic effects for conditions including atopic dermatitis and inflammatory bone loss in vivo [25]. In addition, it inhibits the production of important cytokines in various diseases by reducing NF-κB and MAPK signaling [26]. TLR signaling pathways include activation of NF-κB and MAPK signaling. Its inhibitory effect on NF-κB and MAPK signaling could inhibit TLR signaling; however, only TLR4 inhibitory effect was reported [27]. Based on the experimental results, synthetic compounds, such as the ssRNA (single-stranded RNA) mimic poly(I), were used to induce TLR3 activity and NF-κB-mediated signal transduction pathway activity to identify a therapeutic agent for TLR-mediated diseases. Previous studies have reported results for poly(I) as a TLR3 ligand and shown that poly(I) recognition results in a response similar to that induced by the TLR3 agonist poly(I:C). In this study, OAA significantly decreased expression of IKKα/β and its another gene, including NF-κB in TLR3. The results presented indicate that poly(I)-induced mRNA expression levels of inflammatory cytokines and chemokines, such as MCP-1, IL-1β, IL-8, VCAM-1, and ICAM-1, were significantly reduced by treatment with OAA. We also observed that inhibitory effects through the regulation of IKKα/β activity in the poly(I)-stimulated-TLR3 signaling pathway and its downstream genes of OAA treatment. Therefore, OAA could be a potential therapeutic agent for the management of TLR3-mediated inflammatory diseases.

## 5. Conclusions

Present results support that OAA reduced the expression of several key regulatory genes through IKKα/β suppression in TLR3-mediated NF-κB activation. Although the present study provides new insights into the regulatory effect of OAA via TLR3, it should be further explored for the elucidation of uncharacterized mechanisms of IKKα/β protein, as well as for the development of new anti-inflammatory therapies and functional foods.

## Figures and Tables

**Figure 1 molecules-24-04002-f001:**
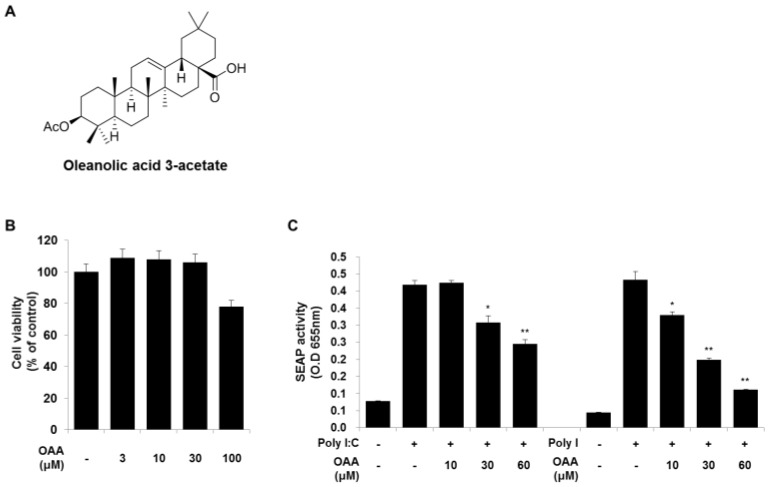
OAA inhibits poly(I)-induced NF-κB/AP-1 activation and cytokine expression in human monocytes. (**A**) Chemical structure of oleanolic acid 3-acetate. (**B**) Cells were incubated with OAA (0–100 μM) for 24 h, and cell viability was determined by the WST-1 assay. (**C**) Cells were pretreated with OAA (0–60 μM) for 1 h before stimulation with poly(I:C) or poly(I) for 18 h (50 μg/mL), and the secretion of SEAP was measured by QUANTI-Blue. Values are presented as ± SD of three individual experiments. * *p* < 0.05 and ** *p* < 0.01 compared with the untreated or Poly I:C and Poly I-treated group.

**Figure 2 molecules-24-04002-f002:**
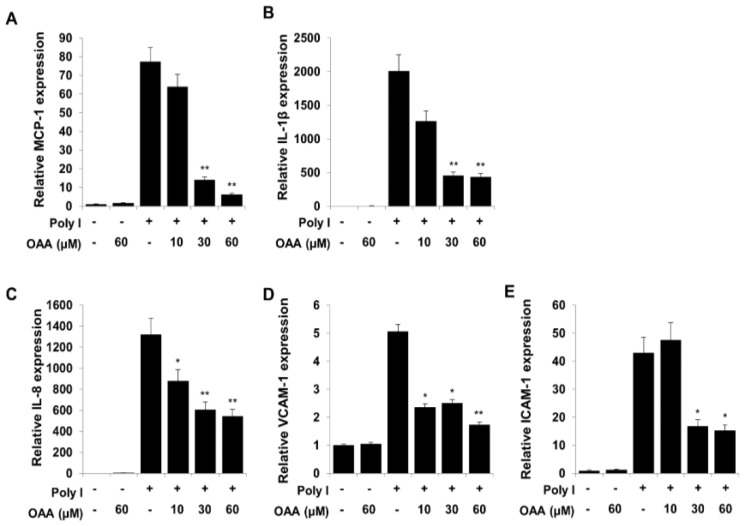
Cells were pretreated with OAA at the indicated concentrations for 1 h before treatment with poly(I) (50 μg/mL) for 12 h, and then total RNA was isolated. The expression levels of inflammatory chemokines or cytokines such as MCP-1 (**A**), IL-1β (**B**), IL-8, (**C**), VCAM-1 (**D**), and ICAM-1 (**E**) were analyzed by quantitative real-time PCR. Values are presented as ± SD of three individual experiments. * *p* < 0.05 and ** *p* < 0.01 compared with the Poly I-treated group.

**Figure 3 molecules-24-04002-f003:**
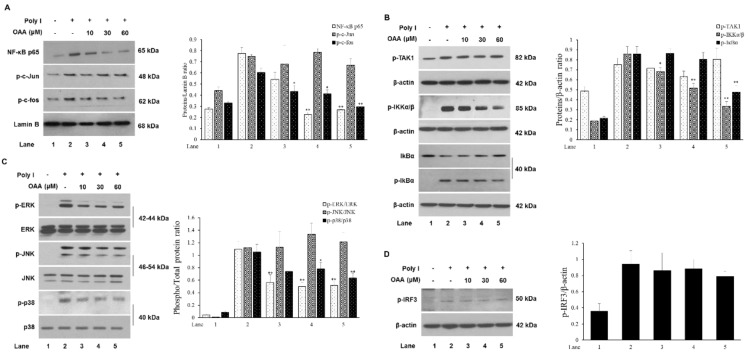
OAA downregulates poly(I)-induced TLR3 signaling. Cells were pretreated with the indicated concentrations of OAA for 1 h and then stimulated with poly(I) (50 μg/mL) for 30 min to 3 h. Cell lysates were prepared for Western blotting analysis with the indicated antibodies. OAA inhibits the poly(I)-induced nuclear translocation of NF-κB and c-fos (**A**), and phosphorylation of IKKα/β, IκB (**B**), ERK, and p38 (**C**) and IRF3 (**D**). Values are presented as ± SD of three individual experiments. * *p* < 0.05 and ** *p* < 0.01 compared with the Poly I-treated group.

**Figure 4 molecules-24-04002-f004:**
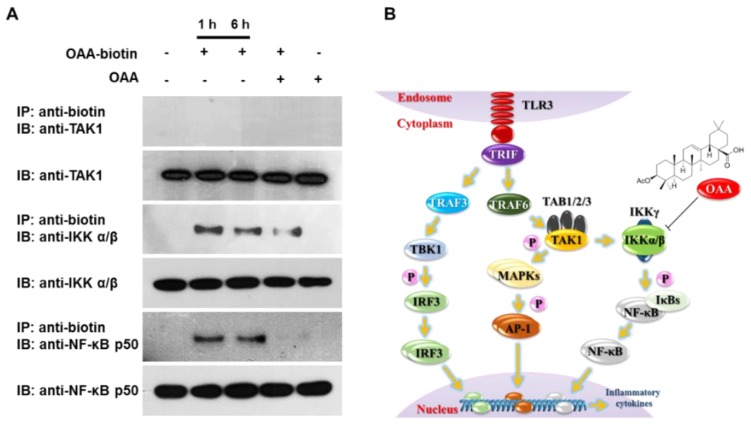
OAA targets IKKα/β. Cell lysates were treated with biotin-OAA or OAA for 1 or 6 h at a concentration of 30 μM, and the lysates were then subjected to pull-down experiments using streptavidin agarose beads. (**A**) The precipitates were analyzed by immunoblotting with the indicated antibodies. (**B**) Proposed model of OAA anti-inflammatory activity in poly(I)-induced TLR3 signaling.

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
