# Peer review of "Oleanolic Acid Acetate Exerts Anti-Inflammatory Activity via IKKα/β Suppression in TLR3-Mediated NF-κB Activation"

_molecules, 2019, doi:10.3390/molecules24214002_

Round 1
Reviewer 1 Report
In this manuscript, Lim et al report that oleanolic acid acetate (OAA) exerts anti-inflammatory effects via suppressing IKKα/β. The authors are requested to address the following issues:
OAA is a derivative of OA. How do these two compounds compare with each other with respect to their anti-inflammatory effects? Does OAA’s acetylation has to do something with its anti-inflammatory effects?
THP cells are not monocytes. Either you should use primary human monocytes or use macrophages. Macrophages can be differentiated from THP-1 cells.
The section on discussion is of least interest. Nothing important is discussed in it. I was expecting that the authors will discuss whether OAA could also inhibit inflammatory responses induced by other TLR agonists.
What is common name of Vigna angularis?
The section on Biotin-OAA makes no sense. There are just so many numbers without any explanation.
Line 131: THP1-XBlue cells are already transfected. You do not need to transfect them with any plasmid.
Compounds 2-5 are described without any context.
Line 315: The sentence is incomplete.
In this manuscript, Lim et al report that oleanolic acid acetate (OAA) exerts anti-inflammatory effects via suppressing IKKα/β. The authors are requested to address the following issues:
OAA is a derivative of OA. How do these two compounds compare with each other with respect to their anti-inflammatory effects? Does OAA’s acetylation has to do something with its anti-inflammatory effects?
THP cells are not monocytes. Either you should use primary human monocytes or use macrophages. Macrophages can be differentiated from THP-1 cells.
The section on discussion is of least interest. Nothing important is discussed in it. I was expecting that the authors will discuss whether OAA could also inhibit inflammatory responses induced by other TLR agonists.
What is common name of Vigna angularis?
The section on Biotin-OAA makes no sense. There are just so many numbers without any explanation.
Line 131: THP1-XBlue cells are already transfected. You do not need to transfect them with any plasmid.
Compounds 2-5 are described without any context.
Line 315: The sentence is incomplete.
In this manuscript, Lim et al report that oleanolic acid acetate (OAA) exerts anti-inflammatory effects via suppressing IKKα/β. The authors are requested to address the following issues:
OAA is a derivative of OA. How do these two compounds compare with each other with respect to their anti-inflammatory effects? Does OAA’s acetylation has to do something with its anti-inflammatory effects?
THP cells are not monocytes. Either you should use primary human monocytes or use macrophages. Macrophages can be differentiated from THP-1 cells.
The section on discussion is of least interest. Nothing important is discussed in it. I was expecting that the authors will discuss whether OAA could also inhibit inflammatory responses induced by other TLR agonists.
What is common name of Vigna angularis?
The section on Biotin-OAA makes no sense. There are just so many numbers without any explanation.
Line 131: THP1-XBlue cells are already transfected. You do not need to transfect them with any plasmid.
Compounds 2-5 are described without any context.
Line 315: The sentence is incomplete.
Author Response
1. OAA is a derivative of OA. How do these two compounds compare with each other with respect to their anti-inflammatory effects? Does OAA’s acetylation has to do something with its anti-inflammatory effects?
=>Thanks for your review. OAA has better anti-inflammatory effect in case of IL-6 mediated inflammation and collagen induced arthritis mice model (Oh et al., 2014).
=>Rheumatology. (2014) 53, 56-64.
2. THP cells are not monocytes. Either you should use primary human monocytes or use macrophages. Macrophages can be differentiated from THP-1 cells.
=>Thanks for your review. We revised manuscript as follow. “Human leukemia monocytic cell line, THP1 …”
=>As you mentioned, using primary human monocytes or THP-1 derived macrophages are more suitable in this research. However, THP-1 cell has been used in many inflammatory researches (Morris et al., 2014, Li et al., 2015, Yang et al., 2016). In further study, we will research OAA could inhibit TLR-3 mediated inflammation via IKKα/β suppression in vivo model and THP-1 derived macrophages (Journal of biological chemistry. (2014) 289, 21584–21590; Acta Biochimica et Biophysica Sinica. (2015) 47, 368–375; Scientific Reports. (2016) 6, 34611.)
3. The section on discussion is of least interest. Nothing important is discussed in it. I was expecting that the authors will discuss whether OAA could also inhibit inflammatory responses induced by other TLR agonists.
=>Thanks for your review. We revised follow your comment in discussion part.
4. What is common name of Vigna angularis?
=>Thanks for your review. Common names of Vigna angularis are red bean, red mung bean and adzuki bean.
5. The section on Biotin-OAA makes no sense. There are just so many numbers without any explanation.
=>Thanks for your review. The section of “Biotin-OAA” presents the NMR spectroscopic data for Biotin-OAA that was first reported in our study. In generally, previously undescribed synthetic or natural chemicals are provided with their physicochemical property such as NMR and HRMS data [Biochemical Pharmacology 89 (2014) 62–73]. It is not a meaningless section, however, “Biotin-OAA section” was transferred to supplementary materials.
6. Line 131: THP1-XBlue cells are already transfected. You do not need to transfect them with any plasmid.
=>Thanks for your review. Line 131 sentence was rived in manuscript as follow: “THP1-XBlue cells are transfected with a reporter plasmid containing SEAP under the control of an NF-κB and AP-1 inducible promoter.”
7. Compounds 2–5 are described without any context.
=>Thanks for your review. As mentioned by reviewer, the description of compounds 2–5 was insufficient in the text and it may be inappropriate for the context. Thus, “Synthetic Procedure for biotin labeling of OAA” has been removed from the manuscript and is included only in supplementary materials so that the text flows smoothly.
8. Line 315: The sentence is incomplete.
=>Thanks for your review. We revised the sentence in our manuscript.
Reviewer 2 Report
The authors present data supporting their idea that oleanolic acid acetate can inflammatory responses initiated through the TLR3 pathway. Overall, this is a well written manuscript, but the following should be addressed.
Lines 124-129 – Section 2.4 Cell culture: It is unclear what the difference is between the THP1-XBlue cells described in the first sentence and the THP1-XBlue cells in the second sentence. Were the cells grown in different antibiotics for a particular reason? What is the reason?
Lines 182-185 – Section 2.10 Statistical Analysis: The data presented in figures 1 – 2 appear to be multiple comparisons to a common control. If so, the student t-test would not be appropriate unless the p values were subjected to a Bonferroni correction. Alternatively, Tukey’s test could be applied.
Figure 1B and 1C – They figures appear to be mislabeled.
Lines 203-207 and lines 217-221 – Figures 1b and 2a-e: It is unclear what group is being compared. Are the comparisons being made to the no poly IC/no OAA control (Figure 1B) and the no poly I/no OAA control (Figure 2A-E) or are they being compared to the poly IC added/no OAA control (Figure 1B) or the poly I added/60 uM OAA control?
Figure 3 – According to the methods, all experiments were performed three times. Would it be possible to quantify the western blot band intensities across the three experiments, at least for a subset of the proteins tested, to support the conclusions? Are the data they presenting in figure 3 representative of the average?
Figure 3 and figure 4 – Can the panels be annotated with the estimated molecular weights of the observed bands, as determined by the molecular weight markers that were presumably run with these gels?
Line 272 – The authors argue that their results demonstrate that OAA interacts with IKKalpha/beta. It may be more appropriate to state their finding “support their idea.” These data were collected in an in vitro system (pull down experiment). Further experimentation using alternative approaches likely will be required to make a definitive conclusion.
Author Response
We appreciate your review of our manuscript entitled “Oleanolic acid acetate exerts anti-inflammatory activity via IKKα/β suppression in TLR3-mediated NF-κB activation. (molecules-619991)”. We also believe that our manuscript has been improved due to your valuable comments and suggestions.
We have revised the manuscript based on the reviewer’s comments and suggestions as follows:
1. Lines 124-129 – Section 2.4 Cell culture: It is unclear what the difference is between the THP1-XBlue cells described in the first sentence and the THP1-XBlue cells in the second sentence. Were the cells grown in different antibiotics for a particular reason? What is the reason?
=>Thanks for your review. THP1-XBlue in first sentence was mistyping of THP1. THP1 and THP1-XBlue cells were maintained different conditions.
2. Lines 182-185 – Section 2.10 Statistical Analysis: The data presented in figures 1 – 2 appear to be multiple comparisons to a common control. If so, the student t-test would not be appropriate unless the p values were subjected to a Bonferroni correction. Alternatively, Tukey’s test could be applied.
=>As suggested by the reviewer, its our mistake and corrected. We used Prism 5 software.
3. Figure 1B and 1C – They figures appear to be mislabeled.
=>Thanks for your review. We revised figure 1.
4. Lines 203-207 and lines 217-221 – Figures 1b and 2a-e: It is unclear what group is being compared. Are the comparisons being made to the no poly IC/no OAA control (Figure 1B) and the no poly I/no OAA control (Figure 2A-E) or are they being compared to the poly IC added/no OAA control (Figure 1B) or the poly I added/60 uM OAA control?
=>Thanks for your review. In figure 1, cell viability data were compared with untreated group and SEAP result was compared with Poly I:C and Poly I treated group. In figure 2, Real time PCR results were compared with Poly I treated group. We revised manuscript and indicated comparison group in figure legend.
5. Figure 3 – According to the methods, all experiments were performed three times. Would it be possible to quantify the western blot band intensities across the three experiments, at least for a subset of the proteins tested, to support the conclusions? Are the data they presenting in figure 3 representative of the average?
=>Thanks for your review. As suggested by reviewer, we quantified western blot band intensities and supplemented it in figure 3. Western blot data in figure 3 represent for the three experiment.
6. Figure 3 and figure 4 – Can the panels be annotated with the estimated molecular weights of the observed bands, as determined by the molecular weight markers that were presumably run with these gels?
=>As suggested by reviewer, we annotated molecular weights of the observed bands of figure 3 and molecular weight of compounds of supplementary data (Figure S1 & S2, MW: 823.5766).
7. Line 272 – The authors argue that their results demonstrate that OAA interacts with IKKalpha/beta. It may be more appropriate to state their finding “support their idea.” These data were collected in an in vitro system (pull down experiment). Further experimentation using alternative approaches likely will be required to make a definitive conclusion.
=>As suggested by the reviewer, we revised manuscript as followed “Present results support that OAA reduced the expression of several key regulatory genes through IKKα/β suppression in TLR3-mediated NF-κB activation.”.
Round 2
Reviewer 2 Report
In the revised text, the authors provided more information about the statistical analyses they performed. Unfortunately, they do not state what statistical tests were chosen. The authors used Prism 5, so they have access to some of the post-hoc tests contained within that software. Did they use Tukey's or some other test, and can they state this explicitly in the methods? Pairwise comparisons are not appropriate in the multiple comparisons they are showing, and if the p values they generated are from t tests, they will need to be Bonferroni-corrected, or re-analyzed.
Author Response
In the revised text, the authors provided more information about the statistical analyses they performed. Unfortunately, they do not state what statistical tests were chosen. The authors used Prism 5, so they have access to some of the post-hoc tests contained within that software. Did they use Tukey's or some other test, and can they state this explicitly in the methods? Pairwise comparisons are not appropriate in the multiple comparisons they are showing, and if the p values they generated are from t tests, they will need to be Bonferroni-corrected, or re-analyzed. =>As suggested by the reviewer, Turkey’s test was performed using Prism 5 software and “Statistical analysis” section of the manuscript was revised as follow:“Statistical analyses were performed 3 times for all experiments. All quantitative results are presented as means ± standard deviations (SD). Statistical analyses were performed using Prism 5 software (GraphPad Software, San Diego, CA) and statistical significance was determined by one-way ANOVA followed by Turkey’s test for multiple comparisons (*p < 0.05).”
